# Predicting Perceived Hedonic Ratings through Facial Expressions of Different Drinks

**DOI:** 10.3390/foods12183490

**Published:** 2023-09-19

**Authors:** Yasuyo Matsufuji, Kayoko Ueji, Takashi Yamamoto

**Affiliations:** Department of Nutrition, Faculty of Health Sciences, Kio University, 4-2-2 Umami-naka, Koryo, Kitakatsuragi, Nara 635-0832, Japan; f1997464@kio.ac.jp (Y.M.); k.ueji@kio.ac.jp (K.U.)

**Keywords:** facial expressions, hedonic ratings, tastant-induced emotions, predictive model, objective assessment

## Abstract

Previous studies have established the utility of facial expressions as an objective assessment approach for determining the hedonics (overall pleasure) of food and beverages. This study endeavors to validate the conclusions drawn from preceding research, illustrating that facial expressions prompted by tastants possess the capacity to forecast the perceived hedonic ratings of these tastants. Facial expressions of 29 female participants, aged 18–55 years, were captured using a digital camera during their consumption of diverse concentrations of solutions representative of five basic tastes. Employing the widely employed facial expression analysis application FaceReader, the facial expressions were meticulously assessed, identifying seven emotions (surprise, happiness, scare, neutral, disgust, sadness, and anger) characterized by scores ranging from 0 to 1—a numerical manifestation of emotional intensity. Simultaneously, participants rated the hedonics of each solution, utilizing a scale spanning from −5 (extremely unpleasant) to +5 (extremely pleasant). Employing a multiple linear regression analysis, a predictive model for perceived hedonic ratings was devised. The model’s efficacy was scrutinized by assessing emotion scores from 11 additional taste solutions, sampled from 20 other participants. The anticipated hedonic ratings demonstrated robust alignment and agreement with the observed ratings, underpinning the validity of earlier findings even when incorporating diverse software and taste stimuli across a varied participant base. We discuss some limitations and practical implications of our technique in predicting food and beverage hedonics using facial expressions.

## 1. Introduction

Taste elicits robust hedonic responses, encompassing sensations of pleasure or displeasure and distinct qualities such as sweetness, sourness, saltiness, bitterness, and umami. It is widely acknowledged that the hedonic aspects of taste trigger emotional responses, ultimately influencing food consumption behavior, consumer preferences, and the acceptance of food products [1]. While the qualitative attributes of taste are primarily assessed through explicit means, involving subjective ratings of perceived taste quality, rather than implicit measurements [2], the hedonic dimension of taste can also be implicitly assessed through physiological reactions, in addition to subjective (explicit) measurements, such as self-reported overall liking using a nine-point hedonic scale. Hence, the combination of implicit and explicit aspects is crucial for the hedonic evaluation of taste [3].

Implicit evaluation methods are diverse [4], encompassing heart rate variability [5,6,7], skin temperature [8], skin blood flow [9,10], skin conductance response [11], facial expressions [3,6,12,13], and brain activity [14,15,16,17]. Among these, facial expression analysis stands out as a relatively straightforward and practical approach, utilizing readily available automatic classifiers for facial affect recognition [18]. In the realm of taste research, Steiner [12] pioneered the revelation that facial expressions occur reflexively in response to basic tastes and can be categorized based on hedonic tone, whether they convey a sense of pleasantness or unpleasantness, rather than specific facial expressions corresponding to the taste’s qualitative attributes. Subsequently, research on taste and facial expressions has made significant advancements.

Numerous studies have examined the facial expressions of individuals while consuming various foods and beverages, extending beyond basic taste solutions [5,6,19,20,21,22,23,24,25,26]. Recent research has placed emphasis on predictive modeling based on these analyses, encompassing the prediction of consumer preferences [27], consumer acceptance of food products [28,29], and even choices in beer selection [25]. However, to date, no studies have sought to predict explicit hedonic ratings by analyzing implicit facial expressions. To address this gap, our research group embarked on a pioneering investigation into the potential of predicting the deliciousness of food and beverages through facial expression analysis in 2021 [30]. In this study, using 10 female students (21–22 years old), our approach utilized facial expression analysis as the dependent variable, quantifying expressions of neutrality, happiness, sadness, surprise, fear, disgust, and anger for each of the five basic tastes (sweetness, saltiness, sourness, bitterness, and umami) in various solutions. As an independent variable, we incorporated participants’ perceived hedonic ratings, obtained through explicit sensory assessments for each taste. Employing multiple regression analysis, we established a regression equation for predicting hedonic ratings. Subsequently, we validated the model by applying facial expression analysis results from different subjects (six females and six males; age range, 22–59) who consumed a range of taste solutions, including commercially available beverages. It is noteworthy that, when the emotional scores of facial expressions from male and female participants (five individuals in their twenties, two in their thirties, three in their forties, and two in their fifties) were input into the prediction formula derived from 21–22-year-old female participants, the predicted values closely matched the perceived ratings for each individual, affirming facial expression analysis as a valuable and objective method for evaluating the hedonic aspects of taste perception.

However, this study identified three limitations: (1) a small sample size; (2) limited generalizability of the AI application used for analysis, which was locally available but not globally accessible; and (3) reliance on a single selected facial expression image chosen by the experimenter (one-shot single image). Therefore, the primary goals of this study were to address these limitations: (1) to increase the sample size to at least double that of the previous study; (2) employ widely used facial expression analysis software, FaceReader, with broader accessibility; and (3) explore the utility of various methods, including data analysis based on average facial expression values over a specific timeframe (in addition to one-shot analysis). The overarching objective was to reaffirm the predictability of food and beverage deliciousness or unpleasantness based on facial expressions with these enhancements.

Foods and beverages encompass a myriad of chemical compounds, and their individual tastes intricately intermingle, rendering qualitative analysis a formidable challenge. Nonetheless, judgments regarding whether something is delicious or not, based on hedonics, can be easily discerned and are readily manifested in facial expressions. Conversely, by leveraging our methodology, which seeks to forecast the degree of deliciousness from facial expressions, we anticipate obtaining quantitative outcomes not only for sensory scientific inquiries at the laboratory level but also for evaluating product preferences, comparing preferences across different products, and expediting preference surveys for new food products.

## 2. Materials and Methods

### 2.1. Participants

In the previous report, a total of 22 participants were used, while in this study, we recruited a total of 49 participants from Kio University, including students and staff members. Based on the responses obtained from a pre-experiment questionnaire, we confirmed that all participants did not have any sensory abnormalities, eating disorders, or mental disorders and were not taking any medications that could potentially affect their sense of taste. We instructed all participants not to eat or drink anything for one hour prior to the start of the experiment. We provided a detailed explanation of the purpose of the experiment, safety measures, and protection of personal information. After obtaining their understanding and consent, we collected written informed consent from each participant. This study received approval from the Kio University ethics committee (No. R2-31), and all experiments were conducted in adherence to the principles outlined in the Declaration of Helsinki.

### 2.2. Experiment 1

The experimental procedure and taste solutions were essentially the same as those of our previous study [30]. A group of 29 healthy female volunteers (age range, 18–55 years; mean ± SD, 23.1 ± 7.9) participated in an experiment aimed at assessing the effectiveness of AI in analyzing facial expressions and establishing a formula for predicting hedonic ratings. Taste solutions were administered to 16 out of the 29 participants and included ten different concentrations of five conventional basic tastes: 2.5%, 5%, 10%, and 20% sucrose; 0.5% and 2% monosodium glutamate (MSG); 1% citric acid; 1% and 5% sodium chloride (NaCl); and 0.01% quinine hydrochloride (QHCl). The remaining 13 participants received taste solutions consisting of 5% glucose, 0.3% sodium guanylate, and 3% NaCl. Each of the solutions was prepared using distilled water (DW). A 10 mL aliquot of the taste solution was placed in a small paper cup positioned in front of the seated participant. Participants were instructed to sip the 10 mL of liquid, hold it in their mouths for approximately 1 s, and then swallow. They were encouraged to display facial expressions naturally without deliberate intent and to provide brief remarks regarding the quality and/or palatability of the stimulus after recognition. Following the consumption of each solution, participants rinsed their mouths with DW. The task was repeated with a minimum inter-stimulus interval of 2 min. The order of stimulus presentation was randomized, except for QHCl, which was administered last due to its lingering taste. Additionally, participants were asked to assess the overall hedonic rating of the stimulus using a scale ranging from −5 (extremely unpleasant) to +5 (extremely pleasant), with 0 indicating a neutral response, prior to commencing the next tasting session. It is worth noting that in our previous paper [30], we employed a scale ranging from −10 to +10.

One researcher was positioned close to the participant and signaled the start of the drinking task while simultaneously recording a video. The video was captured with a digital camera (Cyber-shot DSC-WX350; Sony Corp., Tokyo, Japan) placed 2 m in front of the participant. The participant was instructed to maintain direct eye contact with the camera to obtain a frontal face view. Adequate and uniform white lighting was employed to ensure optimal recording conditions.

After the experiment, the video replay was analyzed using the AI application FaceReader (ver. 8.1; Noldus Information Technology, Wageningen, The Netherlands). The most significant difference between this study and the previous one [30] is the use of FaceReader, whereas a different AI application was employed in the previous report. FaceReader processes facial expressions frame-by-frame at 30 Hz and classifies them into seven emotions (neutral, happy, sad, angry, surprised, scared, and disgusted) with scores ranging from 0 (no visible emotion) to 1 (emotion fully present). We conducted score analyses based on four different methods: (1) a single selected facial expression image chosen by the experimenter (one-shot image) judged to be the most applicable facial expression for the taste stimulation; (2) the average emotion scores for 2 s with the one-shot image in the middle (one-shot ± 1 s, or 2 s image); (3) the average emotion scores for 4 s (one-shot ± 2 s, or 4 s image); and (4) the average emotion scores for 6 s (one-shot ± 3 s, or 6 s image).

Any part overlapping with a subject’s brief remark was excluded from the analysis, and the mean value was calculated from the remaining analysis time.

In the subsequent phase, we conducted a multiple linear regression analysis to predict hedonic ratings utilizing the seven emotions. The calculation was grounded on the scores of the seven emotions collected from 29 participants for 13 stimuli, which served as the dependent variables. The independent variable comprised the participants’ self-reported hedonic ratings for each stimulus. Through this multiple regression analysis, we derived a regression equation for predicting hedonic ratings.

### 2.3. Experiment 2

Another randomly chosen group of 20 healthy volunteers (19 females and 1 male, aged 20–50 years; mean ± SD, 22.9 ± 6.3) took part in a second experiment to assess and validate the effectiveness of the formulae derived in Experiment 1 for predicting hedonic ratings. None of the participants in Experiment 2 had been involved in Experiment 1. The taste stimuli consisted of the following 11 liquids: natural mineral water (ILOHAS, Coca-Cola Bottlers Japan Inc., Tokyo, Japan), 1% malic acid, 2% monopotassium glutamate (MPG), 0.003% sucrose octa acetate (SOA), 7% calorie-free sweetener (Palsweet, Ajinomoto Co. Inc., Tokyo, Japan), peach juice (Peach Mix 100%, Dole Japan, Inc., Tokyo, Japan), noodle broth (Mentsuyu, Daitoku Food Co., Ltd., Nara, Japan), vegetable juice (Thick Vegetable Juice, Kagome Co. Ltd., Tokyo, Japan), 2.5% salt (Hakata-no-Shio, Hakata Salt Co., Ltd., Ehime, Japan), flat lemon juice (Shikwasa juice, Okinawa Aloe Co. Ltd., Okinawa, Japan) and catechin green tea (Healthya Green Tea, Kao Corp., Tokyo, Japan). These stimuli were different from those used in our previous study [30] except for SOA. The taste stimuli were given to the participants randomly, but SOA was given last.

Liquid intake, video recording, FaceReader analysis, and the rating of perceived hedonics followed the same procedures as those utilized in Experiment 1. The FaceReader outputs, representing emotional facial expressions in response to these stimuli, were incorporated into the respective emotion categories within the equations established in Experiment 1, resulting in predicted (or calculated) hedonic ratings. Subsequently, predicted and perceived hedonic ratings were assessed and compared.

Summarizing the research methodology, in Experiment 1, we conducted both perceived hedonic ratings and emotional analysis of facial expressions for basic tastes, leading to the derivation of predictive formulae for hedonic ratings based on multiple regression analysis. In Experiment 2, a different set of participants was given different taste solutions, and we compared the perceived hedonic ratings with the hedonic ratings calculated when including the hedonic scores of facial expressions into the formula. The experimental method was the same as in the previous report [30], but the number of participants, the AI application used, and the analysis method were different.

### 2.4. Data Analysis

In Experiment 1, a boxplot analysis was conducted to assess the scores for the seven emotions associated with each of the 10 stimuli across 16 participants. The analysis yielded median values, interquartile ranges, as well as minimum and maximum scores. To investigate the similarity in hedonics among taste stimuli, Spearman’s correlation coefficients were computed between pairs of stimuli based on the scores of the seven emotions. For a deeper understanding of the relationships between these variables, a multiple linear regression analysis was carried out. In this analysis, the scores for the seven emotions served as the dependent variables, and they were derived from responses to 13 stimuli provided to 29 participants. The participants’ perceived hedonic ratings for each stimulus were used as the independent variables. To check for multicollinearity, which arises when predictors exhibit high correlations, correlation coefficients were computed among pairs of the seven emotions. In Experiment 2, the connections between predicted and perceived hedonic ratings were explored and compared using Pearson’s and Spearman’s correlation coefficients, along with a one-way ANOVA and the Wilcoxon signed-rank test. Before conducting the correlation analyses, data were assessed for normal distribution using the Shapiro–Wilk test. All statistical analyses were performed using IBM SPSS Statistics (ver. 25) and Excel Statistics 2012, with statistical significance set at *p* values < 0.05.

## 3. Results

### 3.1. Experiment 1

The scores for seven emotions, neutral, happy, sad, angry, surprised, scared and disgusted, from the FaceReader outputs for one-shot images taken after the presentation of taste stimuli are shown in Figure 1. The FaceReader output contains ‘contempt’, but in the present study, this term was omitted because contempt is not related with the taste evaluation and no participants showed this emotion to any taste stimuli tested. Figure 1 denotes the hedonic pattern profiles across the seven different emotions in response to 10 taste stimuli in 16 participants. The panels are arranged according to the mean perceived hedonic rating for each stimulus in 16 participants.

Figure 2 depicts the boxplot analysis of the scores corresponding to the seven emotions displayed in Figure 1. It provides a visualization of the median and interquartile range, along with the minimum and maximum scores, for the emotions elicited by the ten stimuli. The profiles of scores for the seven emotions indicate that higher concentrations (10% and 20%) of sucrose exhibit a pronounced happiness component, whereas 1% citric acid, 5% NaCl, and 0.01% QHCl are associated with the largest sadness component. The neutral component predominates for the remaining stimuli.

Figure 3 presents the computed correlation coefficients among the emotional score profiles for the 10 stimuli. Spearman’s analysis was employed due to the non-normal distribution of scores for some stimuli. Statistically highly significant correlations were observed in three distinct groups. The first group included 20%, 10%, and 5% sucrose (Figure 2A–C, respectively), which exhibited highly positive perceived hedonic ratings. The second group consisted of 2% MSG, 0.5% MSG, and 1% NaCl (Figure 2E–G, respectively), showing nearly neutral to slightly negative perceived ratings. The third group comprised 1% citric acid, 5% NaCl, and 0.01% QHCl (Figure 2H–J, respectively), which had highly negative hedonic ratings. Notably, 1% NaCl also displayed strong correlations with 1% citric acid and 5% NaCl in their emotional scores (Figure 2G–I, respectively).

A multiple linear regression analysis was performed to predict hedonic ratings based on the scores of the seven emotions for one-shot images obtained in Experiment 1. In addition to the data obtained in 16 participants, as shown in Figure 1 and Figure 2, the data in 13 participants tested with 5% glucose, 0.3% sodium guanylate and 3% NaCl were combined. Before the analysis, multicollinearity was ensured by examining the correlation coefficients among the seven emotions. No statistically significant (*p* < 0.05) correlation was found for any pair of the seven emotions. As a result, we obtained the following regression formula for one-shot images [F (7, 191) = 3.513, *p* < 0.001 with an adjusted R^2^ of 0.614]:Hedonic rating = 4.914 × surprise + 7.568 × happiness + 0.617 × scare + 4.393 × neutral − 1.402 × disgust − 2.414 × sadness − 3.917 × angry − 3.234,
where happiness (*p* < 0.001), neutral (*p* < 0.01), sadness (*p* < 0.05) and surprise (*p* < 0.05) were significant predictors of hedonic ratings.

For 2 sec images, a regression formula [F (7, 191) = 4.781, *p* < 0.001 with an adjusted R^2^ of 0.474] was obtained for the mean emotion scores:Hedonic rating = 6.701 × surprise + 7.461 × happiness + 1.657 × scare + 3.701 × neutral − 1.776 × disgust − 2.563 × sadness − 3.488 × angry − 3.007,
where happiness (*p* < 0.001), neutral (*p* < 0.05) and surprise (*p* < 0.05) were significant predictors of hedonic ratings, and sadness was marginally significant (*p* = 0.061).

For 4 sec images, a regression formula [F (7, 191) = 5.788, *p* < 0.001 with an adjusted R^2^ of 0.365] was obtained for the mean emotion scores:Hedonic rating = 8.276 × surprise + 7.401 × happiness + 0.452 × scare + 3.343 × neutral − 2.022 × disgust − 2.126 × sadness − 1.155 × angry − 3.008,
where happiness (*p* < 0.001) and surprise (*p* < 0.05) were significant predictors of hedonic ratings, and neutral was marginally significant (*p* = 0.075).

For 6 sec images, a regression formula [F (7, 191) = 6.236, *p* < 0.001 with an adjusted R^2^ of 0.316] was obtained for the mean emotion scores:Hedonic rating = 9.714 × surprise + 7.202 × happiness − 1.935 × scare + 3.284 × neutral − 2.201 × disgust − 1.558 × sadness − 1.268 × angry − 3.030,
where happiness (*p* < 0.001) and surprise (*p* < 0.01) were significant predictors of hedonic ratings.

### 3.2. Experiment 2

The validity of these formulae was examined by applying the obtained emotion scores to another 11 taste stimuli in a different group of 20 participants who had not been exposed to the formulae in Experiment 1. We investigated the correlation between the estimated ratings and the perceived ratings, as well as how well the estimated ratings matched the perceived ratings for each taste stimulus. In four participants, however, there was an apparent discrepancy between the estimated ratings and the perceived ratings, mainly due to the lack of happiness reported for hedonically positive stimuli such as peach juice and noodle stock (Figure 4A). Therefore, the subsequent analyses focused on the remaining 16 participants who exhibited a strong correlation and good agreement between the estimated and perceived ratings (Figure 4B). However, a significant difference was observed between the two sets of ratings for the highly palatable peach juice and the highly aversive SOA (Wilcoxon signed-rank test, *p* < 0.01). This difference in ratings for SOA and peach juice may have been attributed to the limitation of the predicted ratings, which could not reach the maximum hedonic ratings of either −5 or +5. To address this issue, the calculated ratings for peach juice and SOA were multiplied by 1.6. This coefficient was determined based on the ratio between the perceived and calculated ratings for peach juice (4.1/2.6 = 1.58) and SOA (−4.5/−2.7 = 1.67). After this adjustment, the estimated ratings aligned much better with the perceived ratings, with the slope of the regression line improving from 0.697 to 0.891, respectively (Figure 4C).

In Figure 5, the correlation analysis between the mean perceived ratings and mean predicted ratings was depicted, which were calculated using the regression formulae based on the mean emotional scores for different analysis periods: one-shot ± 1 s (2 s image) (Figure 5A), one-shot ± 2 s (4 s image) (Figure 5B), and one-shot ± 3 s (6 s image) (Figure 5C). We utilized Pearson’s correlation analysis since the data for both the calculated and perceived ratings exhibited a normal distribution (Kolmogorov–Smirnov test). Although the correlation coefficient slightly decreased from 0.977 to 0.970 to 0.962 as the analysis period increased from 2 s to 4 s to 6 s images, respectively, the slope of the regression line changed from 0.618 to 0.534 to 0.495, indicating a decrease in concordance between the two sets of ratings with longer analysis time.

In addition to these correlation analyses, we examined the concordance between predicted and perceived ratings for each taste stimulus. The difference between the estimated and calculated ratings for each taste stimulus could serve as an indicator of the level of agreement between the two ratings. The mean difference of ratings for all 11 stimuli was calculated to be 0.606, 0.722, 0.953, and 1.027 for one-shot, 2 s, 4 s, and 6 s images, respectively. One-way ANOVA showed a significant main effect [F (3, 30) = 13.702, *p* < 0.001], and post hoc analysis using the Bonferroni test revealed that the difference in ratings for one-shot images was not significantly different from that of 2 s images but was significantly smaller (*p* < 0.001) than that of 4 s and 6 s images.

Finally, we summarized the time point at which the one-shot image was taken. As depicted in Figure 6, the time point varied depending on the perceived hedonic value of the stimulation, either before or after the participant’s brief remark. One-shot images were more frequently taken after remarks for hedonically positive stimuli such as peach juice and noodle broth, while they were more frequently taken before remarks for hedonically negative stimuli such as 2.5% salt and 1% malic acid. This tendency was statistically significant when Pearson’s correlation coefficient was calculated between the number of one-shot images taken before the remarks and the perceived hedonic values (r = 0.830 and r = 0.973 when SOA was omitted). The same tendency was observed in the data from Experiment 1.

## 4. Discussion

The present study was designed to confirm the validity of our previous findings [30], which indicated that the analysis of facial expressions in response to tastants can predict the hedonic ratings of those tastants. Using one-shot images captured through different AI applications and presenting different taste stimuli to various participants, we obtained consistent results, revealing the following: (1) the five basic tastes could be classified into three hedonic categories: positive, neutral, or negative, based on AI analysis of facial expressions. (2) We established a formula for predicting hedonic ratings using multiple linear regression analysis, considering emotional facial expressions in response to basic taste stimuli. (3) By inputting emotional scores of facial expressions in response to different tastants from different participants into this formula, we found a strong correlation and concordance between predicted (or calculated) and perceived (or subjective) hedonic ratings. Although we should be careful with interpretation because the sample size is still relatively small, these results suggest that a single image of a person’s face can quantitatively predict the extent to which that person enjoys food and beverages.

The function of taste lies in discriminating whether a food item is beneficial or detrimental to the body. Innately, the body possesses a physiological function to find appealing and enhance appetite for items that are good for it, while it naturally finds unappetizing and avoids the intake of items that are detrimental. Therefore, food-related behavior is determined by hedonics, i.e., whether the food is considered delicious or not, and the emotions associated with it. The finding that FaceReader classified the basic tastes into three hedonic categories: positive, neutral, and negative was similarly observed in our previous study [30] using a different AI application. Steiner [12] already stated such hedonic classification based on facial expressions 50 years ago. The analysis of the quality of taste information is also essential as a cognitive function of the brain. It involves storing information about the characteristics of the food, along with its aroma, visual attributes, texture, and more, to be utilized in subsequent eating behavior. This sensory function is necessary for adapting future dietary choices.

For this study, we utilized FaceReader, a widely used, convenient, and accurate automated facial expression recognition system. FaceReader classifies facial expressions into the basic universal human emotions suggested by Ekman and Friesen [31], including happiness, sadness, anger, surprise, scare, disgust, and neutrality. The intensity of these emotions ranges from 0 to 1. Although this software is not fully accurate in its emotion recognition performance [6,18], the analyses of these emotions have been effectively employed in various experimental situations in food research [5,19,20,21,22,27,32]. In our previous study [30], we used a different facial expression analysis software. Comparing two AI applications can be challenging. This is because the AI used in the previous study, unlike FaceReader, is essentially a smartphone’s free app, and detailed information about its algorithm and functionality is not available. Moreover, its functionality is extremely simple: it provides emotional analysis results by uploading a single facial photo (one-shot) to the AI. However, the obtained results of facial expression analysis were remarkably similar (compare Figure 1 and Figure 2 of this study with Figure 1 and Figure 2 of the previous report [30]). While detailed analysis is certainly necessary, it seems that the choice of AI software may not need to be overly meticulous.

In our previous study, the AI application used displayed sadness exclusively rather than disgust emotions for facial expressions induced by aversive taste stimuli, such as 5% NaCl, 0.01% citric acid, and 0.01% QHCl. Only “happiness” was a significant predictor of hedonically positive ratings. In a previous study [30], we posited that these results might be dependent on the AI application used. Facial emotions and scores would be classified differently with different accuracies by a different algorithm [18]. However, as shown in the present study, essentially the same results were obtained by the analysis using FaceReader. The dominant appearance of happiness and sadness in these results may be related to a recent study on an emotion recognition test by Wang et al. [33], who reported that happiness and sadness are unique and independent among the emotions.

FaceReader can provide time course data showing changes in each emotion after tasting. A challenging aspect of facial expression analysis is determining the appropriate time window for analysis, and this approach can vary among researchers. In line with our previous study [30], we selected moments during video observation when we judged facial expressions had changed and conducted FaceReader analysis at those “one-shot” moments. However, we cannot be entirely certain if the chosen moments are the best ones. Therefore, we also calculated the average emotional values within a 2 s window, consisting of 1 s before and 1 s after the one-shot moment, as well as within 4 s (2 s before and 2 s after) and 6 s (3 s before and 3 s after) windows. Satisfactory predictive results were obtained with the 2 s window, but as the window width increased, the predictive accuracy decreased. This variation is because emotional scores fluctuate over time. The fact that a 2 s window is acceptable implies that there is no need to be overly meticulous in selecting one-shot moments.

Although we generally obtained a good correlation and concordance between predicted (calculated) and perceived (subjective) hedonic ratings, a significant difference was detected between the predicted and perceived ratings for the very aversive SOA and the very palatable peach juice. Such a difference between the two ratings for SOA and peach juice may have been due to the limitation of the predicted ratings in reaching the maximum hedonic ratings, such as −5 to +5. This phenomenon was similarly detected in our previous study [30] where we used a scale from −10 to +10. To address this issue, the following procedure may be effective: if the estimated rating for a tastant, whose perceived rating is larger than 4.0 or smaller than −4.0, is multiplied by 1.6, the compensated ratings yield calculated intensities that become very close to the perceived intensity, as proven in the present study (see Figure 4C).

We asked each participant to make a short remark after the oral intake of tastants. An interesting new finding was that one-shot images for aversive tastants tended to appear before the remarks, while those for palatable tastants tended to appear after the remarks. These characteristic differences were proven to be statistically significant. This finding agrees with other studies showing that aversive facial expressions appear more intensely and quickly than pleasant expressions [34,35,36], reflecting that aversive tastes convey warning messages in the form of discomfort, harm, and urgency. It is believed that these rapid reactions are mediated by reflex circuits in the brainstem [12]. Good tastes are pleasant, palatable, and nutritive; we enjoy them slowly, comfortably, and with emotional fulfillment. Reflex reactions for palatable tastants in the brainstem are more likely to trigger responses in the autonomic nervous system and hormonal systems rather than the motor system.

However, the bitter-tasting SOA elicited one-shot images both before and after remarks evenly, despite being the most aversive stimulus. This may be explained by the fact that bitter stimuli stimulate taste cells in the foliate and circumvallate papillae situated at the back of the tongue better than those in the anterior tongue [37,38]. About half of the participants felt a stronger bitter taste after remarks at the timing for the bitter substance to reach the posterior tongue.

There are some limitations in this study: (1) the prediction is not successful for individuals who show no or very small expressions of happiness to very palatable tastants, e.g., peach juice and noodle stock, even though these people can express aversive emotions to unpalatable tastants, e.g., malic acid and SOA. In the present study, 4 participants out of 20 belonged to this category of individuals (see Figure 4A). On this point, Zeinstra et al. [39] reported that facial expressions were suitable to measure dislike, but not for pleasant stimuli in school-aged children. Using other facial expressive measurements such as analyzing facial muscle movements might overcome these shortcomings [4]. (2) It was demonstrated that for extremely delicious or extremely unpleasant beverages, adjusting the calculated values can bring them closer to the participants’ sensory evaluations. Since compensation is an operation that is not ideally desired, investigating the existence of AI software that does not require compensation is one of the potential themes for future research. (3) The participants in this study were predominantly women in their twenties. This raises questions of whether factors such as gender, age, ethnicity, etc., are reflected in the predictive formula for hedonic ratings derived from facial expressions. In a previous report [30], using one predictive formula obtained from young women, it was suggested that it could produce hedonic ratings that closely matched the actual values regardless of gender or age. Therefore, in future research, we would like to confirm these aspects by increasing the number of participants. Additionally, we are interested in exploring what results may arise if the AI used to derive the formula differs from the AI used during testing.

The present study has confirmed the validity of our previous findings, showing that hedonic ratings can be effectively predicted by a formula derived from multiple regression analysis of facial expressions obtained using AI software. Our method of predicting hedonic ratings from facial expression emotion analysis initially requires the derivation of equations. However, in future research, if standardized formulae for each AI application are established, the process of deriving formulae may be eliminated in practical situations. By incorporating these standardized formulae into AI systems, users would only need to specify the analysis point (one-shot) during food and beverage consumption, and the hedonic rating would be immediately displayed. This convenience and speed would enhance work efficiency and enable the retrieval of consumers’ hedonic ratings without relying on traditional subjective evaluation methods like analog scales. In product development and consumer preference surveys, this approach would allow for the use of a wide range of consumers without the need for specialized panelists, and hedonic ratings (overall deliciousness) could be obtained rapidly and conveniently. To achieve this, it is necessary to expedite research for standardizing versatile formulae.

## Figures and Tables

**Figure 1 foods-12-03490-f001:**
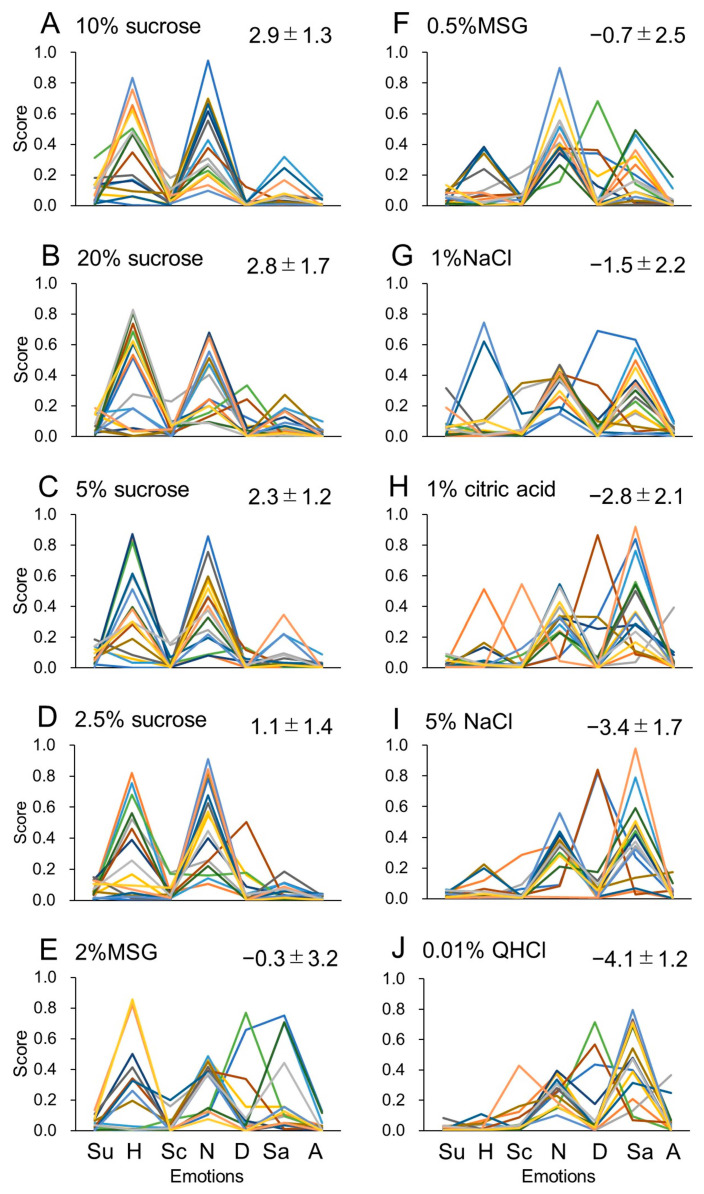
Profiles of hedonic patterns across seven different emotions based on FaceReader analysis of facial expressions in response to 10 stimuli from (**A**–**J**) in 16 participants (shown in different color lines). The analysis was based on one-shot images. Profiles are depicted in different colors for the 16 participants. The emotions are arbitrarily arranged from left to right in the order of surprise (Su), happiness (H), scare (Sc), neutral (N), disgust (D), sadness (Sa) and anger (A). Panels are arranged from the most pleasant to the most unpleasant stimulus from A to J, respectively, as shown by the mean ± standard deviation perceived hedonic rating for each stimulus in 16 participants.

**Figure 2 foods-12-03490-f002:**
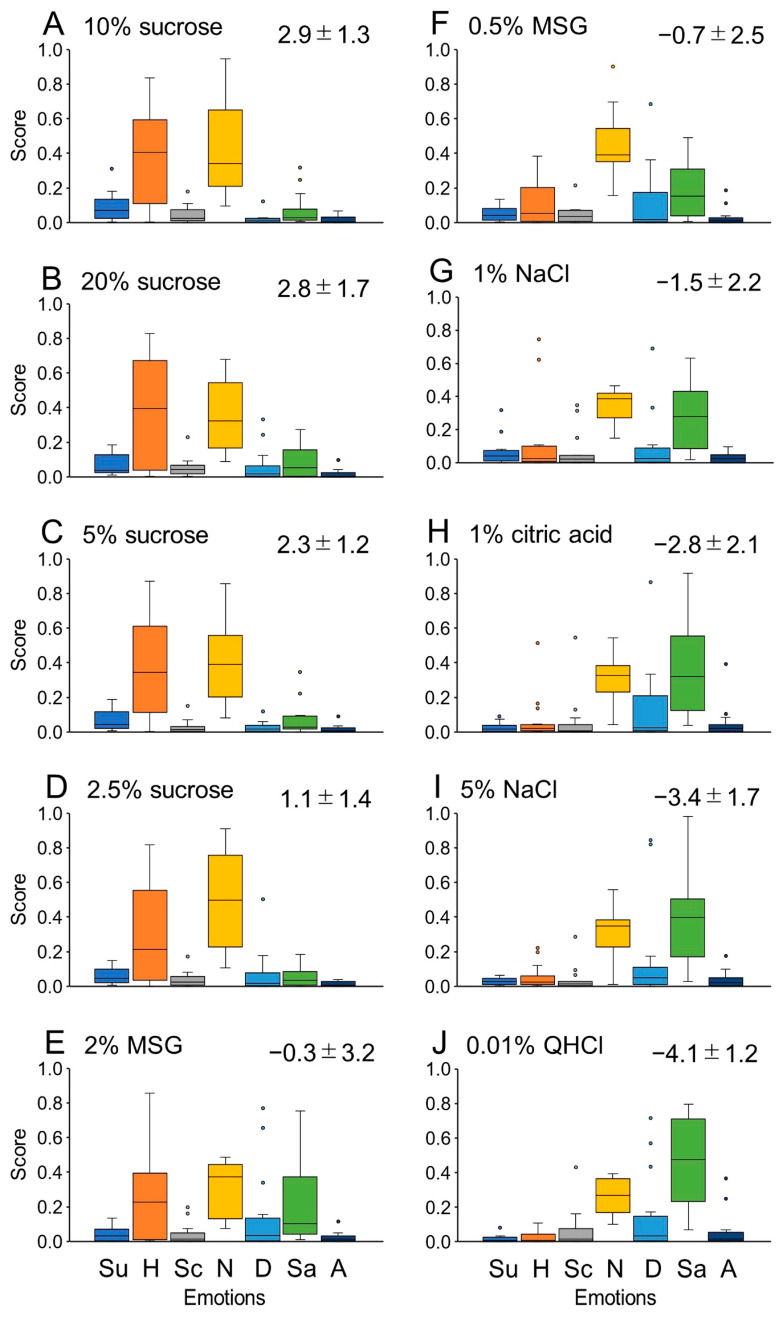
Box and whisker plot analysis of scores for seven different emotions evoked by 10 stimuli from (**A**–**J**). The median and interquartile range with minimum and maximum scores are based on the emotional scores of the 16 participants shown in Figure 1. Small circles in each panel indicate outliers. Other descriptions are the same as those in Figure 1.

**Figure 3 foods-12-03490-f003:**
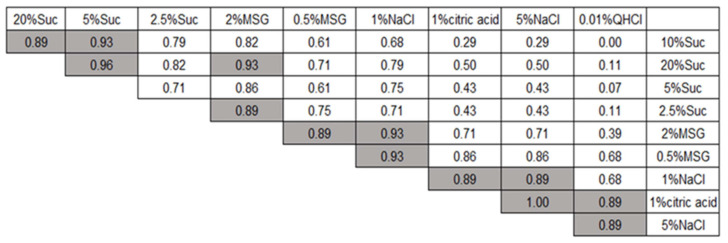
The Spearman’s correlation coefficient matrix of hedonic profiles for the 10 stimuli. Correlations were determined by comparing pairs of stimuli based on the mean scores of the seven emotions depicted in Figure 1. The taste stimuli were organized in descending order of perceived hedonic ratings, from most positive to most negative, both from left to right and top to bottom. Shaded coefficients indicate highly significant correlations (*p* < 0.01).

**Figure 4 foods-12-03490-f004:**
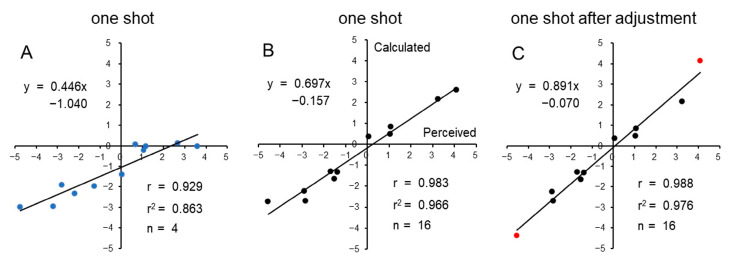
Scatterplots depicting the relationship between mean predicted (calculated) and perceived hedonic ratings for 11 stimuli among 20 participants. The mean calculated hedonic ratings are shown on the y-axis, and the perceived hedonic ratings are shown on the x-axis. Calculation was based on the emotion scores for one-shot images. Each graph includes the regression formula, Pearson’s correlation coefficient (r), coefficient of determination (r^2^), and the number of participants. (**A**): Scatterplots for 4 participants who exhibited very low happiness scores, as indicated in blue. (**B**): Scatterplots for the remaining 16 participants. (**C**): Scatterplots for the 16 participants after the calculated ratings were multiplied by 1.6 for peach juice and SOA, as indicated in red. Following this adjustment, the slope of the regression line is close to 1.0, indicating a significantly improved agreement between the calculated and perceived ratings.

**Figure 5 foods-12-03490-f005:**
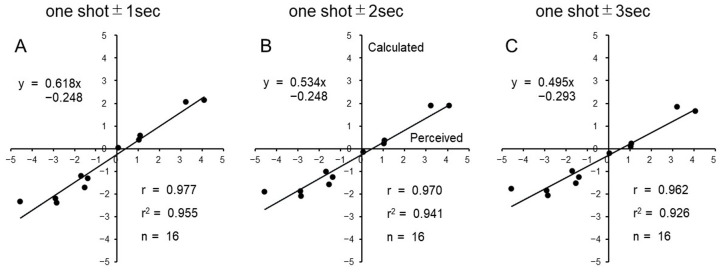
Scatterplots depicting the relationship between mean predicted (calculated) and perceived hedonic ratings for 11 stimuli among 16 participants. The calculated hedonic ratings are displayed on the y-axis, and the perceived hedonic ratings are on the x-axis. Each graph includes the regression formula, Pearson’s correlation coefficient (r), and coefficient of determination (r^2^). (**A**): Calculation based on the mean emotion scores for 2 sec images. (**B**): Calculation based on 4 sec images. (**C**): Calculation based on 6 sec images.

**Figure 6 foods-12-03490-f006:**
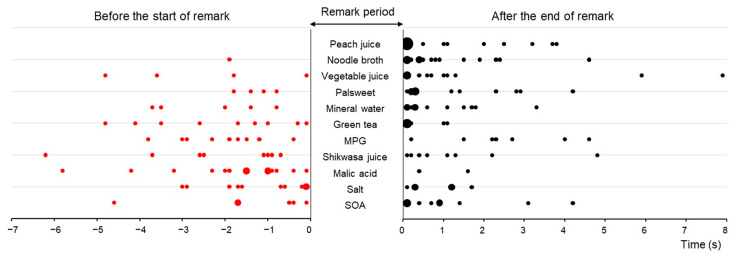
The distribution of one-shot images judged to represent the most relevant facial expression for each taste stimulus. The middle part displays the period of short remarks by participants regarding different tastants, arranged in order of the most palatable to the most aversive from top to bottom. Solid circles indicate the time points of the one-shot images: red circles represent points elicited before the start of the remark, while black circles represent those after the end of the remark. The size of the circle increases depending on the number of occurrences at the same time point. It is noteworthy that points tend to occur after the remark for the palatable stimuli, but before the remark for the aversive stimuli.

## Data Availability

Data are available on reasonable request from the corresponding author.

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
