# Peer review of "Predicting Perceived Hedonic Ratings through Facial Expressions of Different Drinks"

_foods, 2023, doi:10.3390/foods12183490_

Round 1
Reviewer 1 Report
Dear Authors,
I would like to extend my gratitude for granting me the opportunity to read your paper. Your research on the prediction of hedonic ratings based on facial expressions is intriguing and valuable to sensory science and consumer studies. I appreciate the effort you have put into presenting your findings. As I reviewed your work, I identified some areas that could potentially enhance the impact and clarity of your research. I encourage you to consider the suggestions I've provided as ways to strengthen the manuscript's overall quality and relevance. Your dedication to this research is commendable, and I'm confident that your work has the potential to contribute significantly to the advancement of knowledge in this area.
Best regards,
Title:
The title effectively encapsulates the paper's focus on predicting hedonic ratings using facial expressions. However, for further clarity and conciseness, it could be slightly revised.
Suggested Title: "Predicting Perceived Hedonic Ratings through Facial Expressions of Different Drinks"
Abstract:
The abstract conveys the study's intentions, methods, and outcomes clearly. However, there are opportunities to enhance clarity, precision, and language use. Below are suggested improvements:
- Clarity and Precision:
- The abstract is generally well-structured, but a few sentences can be refined for better precision.
- Conceptual Improvement:
- The abstract could benefit from a brief sentence emphasizing the practical implications or significance of the findings.
Suggested abstract:
Suggested Abstract: Previous studies have established the utility of facial expressions as an objective assessment approach for determining the hedonics (overall pleasure) of food and beverages. This study endeavors to validate the conclusions drawn from preceding research, illustrating that facial expressions prompted by tastants possess the capacity to forecast the perceived hedonic ratings of these tastants. Facial expressions of 29 female participants, aged 18-55 years, were captured using a digital camera during their consumption of diverse concentrations of solutions representative of five fundamental tastes. Employing the widely-employed facial expression analysis application, FaceReader, the facial expressions were meticulously assessed, identifying seven emotions (surprise, happiness, scare, neutral, disgust, sadness, and anger) characterized by scores ranging from 0 to 1—a numerical manifestation of emotional intensity. Simultaneously, participants rated the hedonics of each solution, utilizing a scale spanning from -5 (extremely unpleasant) to +5 (extremely pleasant). Employing a multiple linear regression analysis, a predictive model for perceived hedonic ratings was devised. The model's efficacy was scrutinized by assessing emotion scores from 11 additional taste solutions, sampled from 20 other participants. The anticipated hedonic ratings demonstrated robust alignment and agreement with the observed ratings, underpinning the validity of earlier findings even when incorporating diverse software and taste stimuli across a varied participant base.
Keywords: Facial Expressions; Hedonic Ratings; Tastant-induced Emotions; Predictive Model; Objective Assessment
3. What is this “フォームの始まり” (form no beginning)
**Introduction**
The introduction appears to lack depth in terms of contextualizing the research within the broader field of taste perception and hedonics evaluation. While it briefly mentions that taste elicits hedonic responses, it doesn't delve into the significance of understanding these responses for food and beverage industries, health, or even psychology.
Although it highlights the role of facial expressions as indicators of hedonics, it doesn't elaborate on why facial expressions are particularly relevant for this purpose. Further discussion could have been included to explain the physiological and psychological link between taste perception and facial expressions, thus grounding the research in relevant theoretical frameworks.
While the previous study's findings are briefly mentioned, the introduction doesn't sufficiently emphasize their impact on the field. The lack of detail regarding the specific findings or breakthroughs of the previous research might undermine the reader's understanding of why the current study is important.
The limitations of the previous study are outlined, but their potential impact on the validity of the findings isn't deeply explored. A critical analysis of these limitations and their implications could provide a more comprehensive understanding of why improvements are necessary.
Additionally, the introduction could have benefited from a more elaborate explanation of the practical implications of predicting food and beverage hedonics using facial expressions. How might this impact industries, consumer behavior, or even sensory science as a whole?
In conclusion, the introduction lacks depth, critical analysis, and detailed contextualization of the research topic. Expanding on the significance of taste perception, emphasizing the importance of previous findings, critically evaluating limitations, and discussing the broader implications of the study could provide a stronger foundation for the research.
**Materials and Methods**
The materials and methods section provides a comprehensive insight into the experimental design and procedures undertaken. However, some aspects could be addressed for further clarity and methodological rigor.
1. **Participants Selection and Demographics:** The participant selection is well-described, but the sample size appears relatively small (49 participants). Consider discussing the rationale behind the sample size and its potential implications for the study's generalizability and statistical power.
2. **Informed Consent and Ethics:** The section highlights obtaining informed consent and ethical considerations, which is commendable. However, it would be valuable to elaborate on any specific procedures employed to ensure participant understanding and compliance with the research protocol.
3. **Experiment Details:** The section outlines Experiment 1 and Experiment 2 with the stimuli used. While the detailed stimuli are explained, there is a lack of clarity regarding the experimental design. Provide a more explicit outline of the experimental setup, including the order of presentation, counterbalancing, and any potential carryover effects between different taste solutions.
**Discussion**
The discussion provides a comprehensive overview of the study's outcomes, linking them to prior research and offering valuable insights. However, certain aspects could be expanded upon to provide a deeper understanding and address potential questions.
1. **Linkage to Previous Findings:** The discussion effectively highlights the study's aim to validate previous findings. To enhance clarity, consider providing a brief recap of the core findings from the introduction and results sections before delving into their implications.
2. **Categories of Tastes:** The categorization of basic tastes into hedonic categories is intriguing. Elaborate on the theoretical underpinnings of these categories and how they align with previous literature on taste perception and emotion. Discuss any potential implications of this categorization for the broader understanding of taste hedonics.
3. **Utility of FaceReader:** While the study acknowledges FaceReader's utility, further elaboration on its advantages and limitations would be beneficial. Discuss its potential applications beyond the current study and address any concerns about its accuracy or potential biases.
4. **Time Window of Analysis:** The discussion touches upon the time window for analysis of facial expressions. Expand on the rationale behind the different time intervals chosen (1 sec, 2 sec, and 3 sec), and discuss how this information contributes to the understanding of the connection between facial expressions and hedonic ratings.
5. **Characteristics of Aversive Tastants:** The discussion briefly mentions characteristic differences between aversive and palatable tastants. Offer more insights into why aversive tastants tend to generate facial expressions before remarks, while palatable ones tend to elicit expressions after remarks. Explore potential physiological and psychological factors driving these differences.
6. **Limitations and Implications:** The discussion identifies limitations in predicting hedonic ratings via facial expressions. Elaborate on the practical implications of these limitations, including potential applications that might be impacted by these issues. Consider discussing how future research could address these limitations or develop solutions.
7. **Comparison with Previous AI Application:** While the discussion compares the results obtained from FaceReader with those from a previous AI application, more critical reflection on the consistency or differences between these two analyses would provide a richer understanding of the implications of the AI tool choice.
8. **Compensation for Extreme Ratings:** The concept of compensating for extreme predicted ratings is interesting. Provide a more detailed explanation of how this compensation method works and its potential implications for the reliability of predictions.
9. **Broader Applicability:** While the study alludes to potential applications of the research findings, elaborate on the specific contexts where this technique might be applied. Discuss potential benefits and challenges associated with integrating facial expression analysis into consumer surveys or product evaluations.
10. **Conclusion and Research Outlook:** Conclude the discussion by succinctly summarizing the study's main contributions and implications. Consider outlining avenues for future research, such as refining AI algorithms, addressing limitations, or exploring the relationship between facial expressions and other sensory attributes.
Expanding upon these aspects will provide a more comprehensive discussion that not only summarizes the study's outcomes but also enriches the reader's understanding of their significance and potential implications.
4. **Facial Expression Recording:** The description of facial expression recording is clear, but it would be helpful to provide details about the camera's placement, angle, and lighting conditions to ensure consistency across recordings.
5. **AI Application and Analysis:** The utilization of FaceReader is well-justified, but the limitations and accuracy of this software should be acknowledged. Consider discussing any validation studies or known limitations of FaceReader to ensure the reliability of the analysis.
6. **Data Analysis:** The data analysis section is detailed, outlining the steps from emotion scoring to the regression analysis. However, considering the complexity of the analysis methods used, provide a more comprehensive description of the rationale behind choosing specific analysis approaches. Additionally, explain how potential confounding variables were addressed.
7. **Statistical Analysis:** The statistical analyses are appropriately detailed. However, provide more information on the assumptions underlying each statistical test (e.g., normality assumptions) and any steps taken to verify the validity of these assumptions.
8. **Validation and Replication:** While the study mentions addressing limitations from a previous study, elaborate on the specific differences between the current and previous methodologies. This will provide a clearer understanding of how the current study contributes to the existing literature.
9. **Conclusion of Methods:** The section should conclude by summarizing the key methodological steps and the overall experimental design. This will provide readers with a clear overview of the study's methodology.
Incorporating these suggestions will enhance the clarity, transparency, and robustness of the study's methodology section.
**Suggestion for Enhancing the Importance of Implications and Conclusion:**
The implications and conclusion section is a critical component of any research article as it serves to tie together the findings of the study and highlight their broader significance. In the current manuscript, the implications and conclusion section could be enhanced by offering a more in-depth exploration of the practical implications of the research findings, discussing their relevance in a wider context, and outlining potential avenues for future research.
1. **Practical Applications and Relevance:**
The manuscript's findings have the potential to influence various fields, including sensory science, consumer studies, and product development. Detail how the established correlation between facial expressions and hedonic ratings can be practically employed. For instance, emphasize how the proposed technique could contribute to more objective and rapid evaluations of new food and beverage products in the market, potentially saving both time and resources compared to traditional sensory tests. Elaborate on how companies in the food and beverage industry could incorporate this technique to refine their product offerings based on consumer preferences.
2. **Broader Contextual Significance:**
To underscore the broader relevance of the study, discuss how the integration of facial expression analysis into sensory evaluation aligns with the growing emphasis on multidimensional approaches to understanding consumer preferences. Link this study to the broader trend of leveraging technological advancements to gain insights into consumer behavior and perception. Highlight how the study's outcomes contribute to the field's knowledge of the intricate connections between emotion, taste, and preference, shedding light on the fundamental mechanisms underlying hedonic evaluations.
3. **Future Research Directions:**
Engage the reader by suggesting promising avenues for future research based on the study's outcomes. Discuss potential directions that could address some of the current study's limitations, such as improving the accuracy of AI applications for analyzing facial expressions or exploring the impact of cultural variations on the correlation between facial expressions and hedonic ratings. Propose research that could extend the current findings, such as investigating the relationship between facial expressions and other sensory attributes like aroma or texture.
4. **Ethical and Societal Considerations:**
Delve into the ethical implications of using facial expression analysis in sensory research. Discuss the potential benefits and challenges related to privacy, consent, and data security, especially as technology continues to advance. Address how the findings could impact the way studies involving human participants are conducted and monitored.
5. **Conclusion:**
Conclude the implications and conclusion section by summarizing the study's overarching contributions. Emphasize the significance of the research in bridging the gap between emotion and taste perception and its potential to reshape sensory evaluation practices. Reinforce the practicality of the proposed technique and its relevance in enhancing consumer insights. Encourage readers to consider the study's findings in the context of their own research or industry practices.
By expanding the implications and conclusion section along these lines, the manuscript will reinforce the significance of the research findings and provide readers with a comprehensive understanding of how these findings can be applied and built upon in various contexts.
Reviewer 2 Report
The manuscript "Prediction of hedonic ratings of different drinks based on facial expressions" has a topic relevant to "foods" [mdpi]. Although I consider that, globally, I believe that some improvements should be made to the document:
- the abstract should be more detailed with regard to the purpose / objectives of the research,
- the theoretical framework requires greater depth (the manuscript has only 17 references. The manuscript requires further development in the theoretical background
- Authors should make greater efforts to include studies from the last 5 years [Scopus / WoS] to make the manuscript more up-to-date (some studies are more than 20 years old).
- it is suggested that the lines of research for the future and the limitations of the study be developed.
The manuscript "Prediction of hedonic ratings of different drinks based on facial expressions" has a topic relevant to "foods" [mdpi]. Although I consider that, globally, I believe that some improvements should be made to the document:
- the abstract should be more detailed with regard to the purpose / objectives of the research,
- the theoretical framework requires greater depth (the manuscript has only 17 references. The manuscript requires further development in the theoretical background
- Authors should make greater efforts to include studies from the last 5 years [Scopus / WoS] to make the manuscript more up-to-date (some studies are more than 20 years old).
- it is suggested that the lines of research for the future and the limitations of the study be developed.
Reviewer 3 Report
Journal: Foods (ISSN 2304-8158)
Manuscript ID: foods-2575315
Type: Article
Title: Prediction of hedonic ratings of different drinks based on facial expressions
Revision
In the study, the authors used a readaptation of a previous experiment from their own research group to improve the evaluation of a correlation tool between emotions and sensory analysis.
The argument for performing the improvements was the small sample size and limitations of AI applications during the previous research.
As a result of the current study, the authors reported success in predicting satisfaction and emotions, which were well correlated in the form of a regression analysis. The authors also point out the applicability of the method to assess consumer satisfaction with beverages.
According to recent publications (https://doi.org/10.3390/beverages5040062) there is still a deficiency in the assessment of artificial intelligence for a valid analysis of sensory characteristics of beverages.
The current study is therefore relevant to fill these gaps in the use of AI in the sensory analysis of beverages, however the study still needs to emphasize the applicability of the method by making a better comparison of what already exists in the literature and projecting the applicability of the proposed method.
I suggest that the authors add a paragraph in the discussion sections with real proposed applications of the method.
In addition, the article needs other minor points to be revised:
*Please check the texts written in Japanese throughout the draft
Page 2 line 58. Participants are students and staff of Kio University. Is it relevant if the people used in the sample are in an academic environment? Is the level of information or knowledge (in case of students) relevant to obtaining adequate data?
Page 2 line 69. If possible, the authors can simplify the method description by mentioning that experiment 1 was based on previous research (add citation) with modifications and clearly presenting the modifications from the previous study (for example in sample size etc.).
Page 2 line 74. “for 16 of 29 participants”. should it be mentioned in the manuscript how the total n of participants was defined? Why the women were chosen. Why 16 out of 29 participants etc.
Page 3 line 90. “recorded a video focusing”. Did the participants know they were filmed? If not, how were they filmed to avoid influencing their reactions. This is also relevant to be mentioned in the discussion part, as the fact that participants are blind to the assessment and recording methods can interfere with their emotions and reactions to pleasant and unpleasant sensory.
Page 3 line 91. units are sometimes abbreviated like meters (m), sometimes not abbreviated like minutes and seconds. please check if it is desired or in the format for the journal.
Page 3 line 114. “Another group of 20 healthy volunteers” For each experiment, how were participants selected? randomly?
page 3 line 98. If the method relies on some professional skill to select a single shot or moment of emotional expression may have implications for its applicability. Should that fact be commented on in the discussion section?
Page 5 and 6. Figures 1 and 2. The x-axis should have a legend for each graph. it is difficult to correlate the results with each emotion tested.
Page 11 line 380. Shouldn't this paragraph be added as a conclusion section?
English should be revised again. There are many terms that can best be described.

English should be revised again. There are many terms that can best be described.
Round 2
Reviewer 1 Report
All changes are satisfactory.
Author Response
Thank you very much for accepting the manuscript. We appreciate you again for the valuable and kind suggestions, which have been very helpful in improving the manuscript.
Reviewer 2 Report
Although the manuscript (new version) incorporates some of the suggestions I presented, and despite the manuscript now including 39 references, I consider that it would be essential for there to be an "update" considering articles / manuscripts from the last 5 years (Scopus / WoS) in instead of manuscripts over 20 years old. This is a suggestion that I consider important to be guaranteed.
Although the manuscript (new version) incorporates some of the suggestions I presented, and despite the manuscript now including 39 references, I consider that it would be essential for there to be an "update" considering articles / manuscripts from the last 5 years (Scopus / WoS) in instead of manuscripts over 20 years old. This is a suggestion that I consider important to be guaranteed.
Author Response
Thank you very much for the comment on references cited. We have updated the selection of references: 17 out of 39 references were published within the last 5 years and may cover almost all the necessary references for the present research. Old papers published over 20 years ago were cited because of the pioneering research in the relevant field.